# Challenges for Optimization of Reverse Shoulder Arthroplasty Part II: Subacromial Space, Scapular Posture, Moment Arms and Muscle Tensioning

**DOI:** 10.3390/jcm12041616

**Published:** 2023-02-17

**Authors:** Stefan Bauer, William G. Blakeney, Allan W. Wang, Lukas Ernstbrunner, Jocelyn Corbaz, Jean-David Werthel

**Affiliations:** 1Service d’Orthopédie et de Traumatologie, Chirurgie de l’Épaule, Ensemble Hospitalier de la Côte, 1110 Morges, Switzerland; 2Medical School, University of Western Australia, 35 Sterling Highway, Perth, WA 6009, Australia; 3Department of Orthopaedic Surgery, Royal Perth Hospital, Perth, WA 6000, Australia; 4Department of Orthopaedic Surgery, Royal Melbourne Hospital, Parkville, VIC 3050, Australia; 5Department of Biomedical Engineering, University of Melbourne, Parkville, Melbourne, VIC 3010, Australia; 6Service d’Orthopédie et de Traumatologie, Centre Hospitalier Universitaire Vaudois, 1011 Lausanne, Switzerland; 7Service d’Orthopédie et de Traumatologie, Hôpital Ambroise Paré, 9 Avenue Charles de Gaulle, 92100 Boulogne-Billancourt, France

**Keywords:** abduction, subacromial space, scapula, scapulothoracic, lateralization, biomechanics, rotator cuff length

## Abstract

In part II of this comprehensive review on the optimization of reverse shoulder arthroplasty (RSA), we focus on three other challenges: 1. “Conservation of sufficient subacromial and coracohumeral space”; 2. “Scapular posture”; and 3. “Moment arms and muscle tensioning”. This paper follows a detailed review of the basic science and clinical literature of the challenges in part I: 1. “External rotation and extension” and 2. “Internal rotation”. “Conservation of sufficient subacromial and coracohumeral space” and “Scapular posture” may have a significant impact on the passive and active function of RSA. Understanding the implications of “Moment arms and muscle tensioning” is essential to optimize active force generation and RSA performance. An awareness and understanding of the challenges of the optimization of RSA help surgeons prevent complications and improve RSA function and raise further research questions for ongoing study.

## 1. Introduction

In the last decade, our understanding of reverse shoulder arthroplasty (RSA) has progressed significantly thanks to the introduction of software for three-dimensional (3D) CT analysis and preoperative 3D planning. This has enabled surgeons to plan and predict glenohumeral impingement-free rigid body motion with a fixed scapula [1]. However, despite continuous efforts to improve implant positioning, a subset of patients continues to have a poor range of motion (ROM) postoperatively. In some cases, this may be related to scapular posture and thoracic kyphosis [2], as these factors are not yet taken into account in commercially available planning software.

The lateralization of RSA has been demonstrated to increase impingement-free ROM, the deltoid wrapping angle (DWA) and stability of RSA [3,4]. Nevertheless, the optimal positioning of the rotator cuff humeral insertion after RSA remains unclear. It seems logical to aim for the restoration of optimal rotator cuff tension by restoring the anatomical length of the remaining cuff tendon–muscle units in order to optimize function and to prevent overstuffing [5,6,7]. Arm lengthening [8] as well as lateralization [9] will at first improve deltoid fiber pretension and stability; however, care should be taken not to lengthen excessively as muscle overdistension may lead to decreased muscle contraction and nerve injury [10,11]. Biomechanically, RSA with a medialized center of rotation (COR) improves the deltoid moment arm and may be advantageous for force generation but may decrease the aforementioned fiber length and therefore contractile force.

The goal of part II of this review is to make surgeons aware of the significant impact of the subacromial and coracohumeral space and scapular posture on passive and active RSA function and how to assess and address these challenges. As a further essential contribution, part II aims to contribute to the understanding and awareness of the balance between RSA biomechanics and biomechanical force generation on one side of the equation and anatomical muscle length and function on the other side. 

## 2. Conservation of Sufficient Subacromial and Coracohumeral Space

In recent years, it has become understood that, beside all efforts to reduce impingement at the scapular pillar with the arm adducted at the side, it is important to maintain sufficient subacromial and coracohumeral space. Humeral distalization is measured using the acromio-humeral distance (AHD) from the inferolateral acromion to the greater tuberosity [12]. The distance from the glenosphere (GS) to the coracoid and acromion may be reduced using a large GS and the lateralization of the COR. A reduction in this space may coincide with reduced glenohumeral abduction (ABD) and rotation in ABD [13]. Clinically, especially in ABD, scapulothoracic motion has the potential to compensate for reduced glenohumeral motion. However, there are some limitations to the amount of scapulothoracic compensation and, for some patients with scapulothoracic pathology, this compensation can be significantly reduced. It seems logical that RSA should aim to come close to physiological glenohumeral ABD.

Seebauer et al. were the first to analyze clinical ABD after the implantation of the Delta III RSA [14]. They performed dynamic fluoroscopic radiographs of active glenohumeral ABD which was limited to a mean maximum of 53°. A similar limitation of the mean glenohumeral abduction–adduction (ABD-ADD) ROM of 67–25° was reported by Nyffler et al. for the Delta III RSA with a 36 mm GS after implantation according to the manufactures recommendations [15]. When implanted in an inferior position with the baseplate flush at the glenoid rim, the mean ABD-ADD ROM increased to 81–1°.

Gutierrez et al. presented several studies [16,17,18] analyzing the arc of motion for ABD-ADD, establishing a hierarchy of the factors improving ABD under the acromion (Figure 1) [18]. The first factor, factor (1), with the largest effect on ABD, was shown to be the “lateralization of the COR of the GS”, which increases the superomedial space over the GS. This space allows for increased ABD before the impingement of the superolateral rim of the humeral insert with the superior glenoid surface occurs. The second largest substantial effect on ABD was created through the distalization of the GS. It is important that surgeons understand how they can achieve this: Firstly, the goal is to position the baseplate as low as possible, flush with the inferior rim of the glenoid and perpendicular to the supraspinatus fossa line, correcting the RSA angle, as described by Boileau and colleagues [19]. Secondly, a smaller BP is placed flush with the inferior glenoid, which distalizes the GS more (for example 25 mm over 29 mm), creating increased subacromial space and inferior overhang. Thirdly, GS eccentricity may be added, which has the same effects. 

The third largest effect on ABD was derived from the inferior tilt of the GS in the model by Gutierrez, which is equivalent to the correction of the RSA angle as described by Boileau. Inferior tilt should not be overcorrected since this may lead to impingement on the scapula. The fourth largest effect was achieved by changing the neck shaft angle (NSA) from 130° to 150° and the fifth factor consisted of an increase in GS size with a rather minimal increase in ABD from a 36 mm GS to a 42 mm GS. 

An important fact that needs to be remembered from this early study is that a 42 GS does not improve ABD very much compared to the other factors; moreover, if it is positioned too superiorly on the glenoid or with superior inclination, it decreases glenohumeral ABD [18,20].

In summary, the four most important factors to increase ABD and the ABD-ADD range, in hierarchical order, are:(1)Lateralization of COR of GS;(2)GS distalization (BP low, BP small, GS eccentric);(3)GS inferior tilt;(4)Increasing NSA from 130° to 150°.

In a computer model study with a 145° onlay stem design, Lädermann et al. analyzed the notching and glenohumeral ROM of different glenoid configurations [20]. A centered GS showed poorer results for ADD, external rotation (ER), and extension than a lateralized or eccentric GS. A larger GS showed the smallest acromiohumeral distance (AHD), which translated into a larger ABD deficit. The best glenohumeral ABD was seen with a small (36 mm) GS combined with a 10 mm lateralization of the baseplate, in keeping with the aforementioned findings by Gutierrez et al. (Figure 1C) [18].

Lädermann and colleagues did not test lateralization in combination with GS eccentricity in their study, which would have achieved an increased inferior overhang of the GS. The combination of adequate lateralization (3–6 mm) with GS eccentricity has been shown to achieve the best glenohumeral external rotation, extension, and global ROM in a computer model [21]. In Lädermann’s study, glenohumeral ER with the arm at 90° showed a large deficit for the bigger GS (42 mm). In a subsequent study Lädermann showed that the maintenance of sufficient subacromial space is needed to grant sufficient glenohumeral rotation at 90° of ABD [13]. These studies, however, are based on computer modeling, and do not consider the potential of scapulothoracic compensation in vivo.

Sufficient subacromial space can be obtained by distalizing the GS according to Gutierrez’s principles (Figure 1C). Increased eccentricity (e.g., +4 mm instead of +2 mm) distalizes the GS, creating space under the acromion (Figure 1C, green oval) and inferomedially below the scapular pillar (Figure 1C, yellow oval). However, apart from its value for ADD, ER, and EXT, lateralization is the most important factor for increased ABD, as shown by Gutierrez et al. [18]. Bony or metallic baseplate lateralization combined with a larger eccentric GS may apply eccentric load on the baseplate fixation with an increased lever arm, which may not be a biomechanically sound option. GS size should be chosen according to patient habitus and body height, preventing lateral, subacromial, coracohumeral and inferior overstuffing, and to optimize compression and the joint reaction forces of the RSA to avoid instability. For lateralized RSA designs, the use of a large GS (>40 mm) should be limited to bigger patients with outliers of glenohumeral size. 

In summary, despite the fact that most studies are based on glenohumeral ROM analysis without the integration of scapulothoracic ROM, it seems logical that the target for impingement-free glenohumeral ABD for RSA should be close to the physiologic glenohumeral ROM. Conserving the subacromial space is an important means to achieve this aim.

## 3. Scapular Posture

Shoulder mobility may benefit hugely from scapulothoracic movement, with orientation and positioning of the scapula depending on the alignment and posture of the thoracic spine. Kibler summarized the findings of several studies, breaking up shoulder motion into scapulothoracic and glenohumeral motion at a ratio of 1:1 to 1:4 depending on the phases of ABD [22]. For the entire ABD motion, the approximate 1:2 ratio holds true. Therefore, scapulothoracic ROM is of major importance for normal shoulder function. The influence of scapular posture on ROM in RSA has long been overlooked, despite the well-known modifications of the scapula position with aging, due to its intricate dependence on thoracic posture and kyphosis [23,24,25,26]. With advanced age and thoracic kyphosis, the scapula is often located in a more anterior, internally rotated, and protracted position (Figure 2). This leads to modified scapula kinematics and is a known cause for a decrease in overhead ROM of native shoulders [24,27,28,29].

Moroder et al. were the first to study scapular positioning in the setting of RSA based on computer simulation of 100 patients. They classified posture and progressive thoracic kyphosis, scapular protraction, and internal rotation (IR) into three types: A (normal), B (moderate kyphosis, scapular protraction, and rotation), and C (severe kyphosis, protraction, and rotation) (Figure 2). The logical assumption was made that the humeral torsion and GS version should match up, leading to a humeral insert being well aligned and opposed to the GS in the neutral arm position [2]. To achieve this, they recommended adapting the stem’s retrotorsion to match the scapula position to restore an equilibrium between ER and IR, since a protracted and internally rotated scapula may lead to a posterior humeral insert position on the GS (forearm neutral), causing early posterior impingement in ER (Figure 3C). Limitations of the study included the supine position during CT investigations and the nature of a computer simulation without clinical controls.

Reintgen et al. published a retrospective study of a prospective shoulder database with the inclusion of 305 RSA with a mean follow-up of 3.9 years [30]. The influence of thoracic kyphosis on a complete set of ROM parameters after surgery, clinical scores, and radiographic notching were analyzed. Increased kyphosis was seen in females with associated heart disease. There was no influence of thoracic kyphosis on postoperative ROM, clinical scores, or notching. Equal improvements of ROM for all patients were recorded in this clinical study, in contrast to Moroder’s theory based on computer simulation. 

In a second CT-based computer modelling study utilizing preoperative planning software, Moroder et al. examined multiple options of RSA optimization for patients with C-type kyphosis after the initial planning of 30 cases by three experienced surgeons [31]. In comparison to type A patients, type C patients had moderately reduced ADD and ABD and an important reduction in ER and EXT. The authors concluded from their computer simulation that, in C-type patients, higher retrotorsion, a lower neck shaft angle, and a larger or inferior eccentric GS seem to be advantageous.

It is important to recognize patients with severe thoracic kyphosis and internally rotated and protracted scapular posture, as shown in Figure 3, since they may be vulnerable to anterior overstuffing (Figure 3B) with a relative contraindication for RSA lateralization. It may be advantageous to adapt implantation strategies for these patients. The scapular rotation and protraction illustrated in Figure 3 may well have been best treated using “Combined Retroversion” (GS and humerus) to a total of more than 30–40°.

## 4. Moment Arms and Muscle Tensioning

Sufficient lateralization (LAT) on the glenoid leads to decreased notching [32,33] and increased impingement-free ROM [16,20], albeit with the drawback of a decreased deltoid moment arm in ABD and elevation (Figure 4) [5,9]. However, excessive lateralization may cause acromion and scapular spine stress fractures [34,35,36], and the glenoid baseplate may be exposed to higher shear forces, which can lead to early loosening [37]. Sufficient global LAT (glenoid sided, humeral sided, or both combined) may be beneficial to increase the tension and function of the remaining rotator cuff [6,38] and to restore the deltoid wrapping angle (Figure 4B,C), which increases joint reaction forces [39], subsequently increasing RSA stability [6]. The lateral humeral offset (LHO), which is the distance between the medial edge of the coracoid process and the greater tubercle, is considered to be an important parameter to restore, regardless of whether it is achieved on the glenoid or humeral sides [40].

Humeral-sided LAT can be achieved using a curved stem [13], onlay system [9], or by shifting the humerus laterally by dialing and positioning the long distance of the onlay eccentricity medially [4]. The position of the stem, tray, and insert can be described as “onlay” if the tray is positioned on the anatomical neck osteotomy, “semi-inlay” if half of the tray is positioned distal to the neck osteotomy, and “inlay” if the tray or stem is completely inset in the metaphysis distal to the anatomical osteotomy. Very highly lateralized (VHL) implants lead to tuberosity lateralization beyond the anatomic position [3] and may result in problematic overstuffing in smaller patients [19] with resultant poor motion, polyethylene wear [5,6], difficulties reducing the humerus, nerve traction, pain, an inability to repair the subscapularis [38,41], and acromial impingement. The interindividual humeral head diameter has been studied and documented to have a range of difference of 17 mm [37]. With an RSA implant system, this range of the variation of the glenohumeral size may need to be modular to adapt LAT to an individual patient’s anatomy.

Lädermann et al. published a computer model study based on healthy shoulder anatomy with the aim to report on “the best glenoid configuration in onlay reverse shoulder arthroplasty” [20]. They used a constant 145° onlay design and compared four different variations of implanting a 36 mm GS including 10 mm BIO-RSA lateralization as well as a 42 mm concentric GS, all with a 29 mm BP. They measured the length changes of the rotator cuff for each glenoid implantation strategy, dividing the rotator cuff into supraspinatus, three anterior subscapularis units, and three posterior infraspinatus units (upper, middle, lower). They concluded that the variation of the GS configuration leads to ROM and muscle length changes after RSA, and that a 36 mm eccentric GS optimizes ROM whilst limiting scapular notching. The analysis of their data on cuff length changes shows without exception that the inferior and the anterior cuff is less vulnerable to overtensioning than the superior, anterosuperior, and posterosuperior rotator cuff. The posterosuperior cuff of a nonarthritic shoulder was most overtensioned with a 36 mm GS with 10 mm BIO-RSA. This is consistent with the findings of a previous study by Giles and colleagues, who showed the antagonistic potential of the rotator cuff in RSA, which may contribute to increased joint compression and stability, albeit with a risk of overstuffing and deleterious effects [7].

Werthel et al. recommend the positioning of the greater tuberosity of the humerus after the implantation of RSA to ±0 mm compared to the healthy preoperative state [3]. The authors point out that this is best achieved by using RSA implants classified as highly lateralized (HL). One can conclude that RSA lateralization, with its advantageous effects of decreased notching, increased ROM, deltoid wrapping angle, and stability needs to be balanced with the tension of the posteroinferior cuff. Depending on the RSA design, the amount of lateralization, and distalization, the superior and posterosuperior rotator cuffs may act as antagonists. It is therefore important to balance lateralization and distalization, aiming to preserve the posterosuperior cuff even though it is frequently necessary to release the supraspinatus tendon, if still intact, to avoid excessive tension and difficulties in reducing the RSA.

### 4.1. Biomechanics: Moment Arms and Muscle Tensioning

Lateralization of the glenoid leads to a decreased deltoid moment arm in ABD and elevation, as shown in Figure 4B [9], since the COR moves closer to the deltoid line of pull, therefore increasing the force required for ABD [5]. There are many ways to measure lateralization radiographically. Boutsiadis et al. quantified it on postoperative anterior–posterior radiographs using the lateralization shoulder angle (LSA). (Figure 5A) LSAs between 75° and 95° showed a positive correlation with active external rotation, constant score, and activities of daily living requiring external rotation (ADLER) scores [42]. Mahendraraj et al., however, demonstrated the marginal correlation of LSA measurements with clinical outcomes [43]. Neeley et al. used humeral distalization (AHD) measured from the inferolateral acromion to the greater tuberosity, and glenohumeral offset (GHO) was measured from the glenoid face to the greater tuberosity to compare onlay and inlay RSA designs [12]. They demonstrated no significant difference in GHO or AHD between the two groups as their implantation techniques were adapted to achieve soft tissue tension, with many of the inlay designs being implanted in an “onlay” manner, with the humeral tray placed above the level of the humeral osteotomy. 

One can distinguish three main types of RSA descriptive for medial (M) or lateral (L) glenoid (G) and humeral (H) lateralization: (1) MGMH, (2) LGMH, (3) MGLH. Hamilton et al. posited that, in theory, an improved moment arm for the posterior deltoid in ER at 30° of ABD can be expected, as shown in Figure 4C and Figure 6c,d. The MGLH design was found to increase the efficiency of the posterior deltoid relative to the other two designs. While the relative increase in the anatomic moment arm is substantial (60% increase in efficiency relative to anatomic), Figure 6d illustrates the large disparity between the magnitude of the rotation moment arm of the posterior deltoid and the infraspinatus and teres minor, which are the primary external rotators in the native shoulder. The decreased ER seen in RSA may be exacerbated by reverse shoulder designs with medial humeral stem positions, because the already small posterior deltoid ER moment arm is further decreased relative to the anatomic shoulder. 

In contrast to a biomechanical approach used to optimize the moment arm for ER, Di Giacomo has presented the results of a computer model in January 2022 showing improved deltoid fiber recruitment for ER at 60° of ABD with an RSA lateralized at the glenoid (LGMH) compared to RSA lateralized at the humerus (MGLH). He postulated that the rotator cuff fiber length is closer to the normal anatomy for LGMH RSA, which may be advantageous for the contractility of the muscle fibers of the cuff and deltoid according to the Blix curve [10], as well as advantageous for active ER due to fiber recruitment [44]. To date, it is unknown how to combine the biomechanical approach of the optimization of the moment arm and forces with the anatomical RSA approach aiming to optimize fiber recruitment and the length of contractile elements according to Blix. A challenge for the research and development of 3D planning for RSA in the next years will be the incorporation and planning of the length of the musculotendinous rotator cuff units compared to the preoperative and native anatomical state with the help of artificial intelligence. With currently available software, it is possible to plan the change of the lateralization and distalization of the humerus after the implantation of RSA. However, the exact change of the length of the rotator cuff units remains unknown. Furthermore, in patients with long-standing degenerative arthritis, the posterior cuff and muscles may be degenerate and stiff, affecting assumptions of muscle fiber elasticity.

As has been recommended in recent years, a combination of glenoid-sided and humeral LAT with an aim to position the greater tuberosity at ±0 mm of the anatomical humerus position (position in Figure 4D) is the approach we use [3]. 

### 4.2. Arm Length and Prevention of Nerve Injuries

Arm length is an important parameter for RSA stability in the Grammont design prior to the trend of lateralization with an increased deltoid wrapping angle, cuff tension, and stability. Lädermann and colleagues conducted a study of RSA with the Grammont design, assessing arm and humeral length with standardized bilateral radiographs [45]. After RSA, they found a mean change of arm length of 23 mm (S.D. 12, range 1–47). There was no correlation between lengthening and acromial stress fractures and neurological complications; however, there was a statistically significant relationship between shortening and instability. Planning on bilateral comparative standardized radiographs was recommended for challenging cases. In 2013, the same senior authors conducted a systematic review of arm lengthening in RSA, including seven studies [8]. They concluded that the restoration of the humeral length and an increase in the acromiohumeral distance (adequate deltoid tension) is critical for RSA function and stability, citing an arm lengthening distance from the anatomic state of up to 20 mm as a reasonable goal. Boutsiadis reported on the value and clinical correlation of the distalization shoulder angle (DSA). A DSA between 40° and 65° was correlated with better active elevation and abduction. There was a negative correlation between the DSA and LSA (Figure 5), showing that highly lateralized RSA has a high LSA and low DSA and vice versa [42].

We point out that overlengthening (distalization) from the anatomic state may overtension the deltoid, and that, most importantly, the upper subscapularis and upper posterior rotator cuff may lead to nerve injuries, as described by Marion and colleagues [11]. They showed that distalizing the summit of the humerus below the midline of the glenoid (equator) stretches the nerve the most, more so than the lateralization of the humerus. Surgeons should also be aware that posterior–inferior retraction of the humerus during glenoid preparation and the implantation of the GS [11], as well as external rotation, adduction, and extension [44] during humeral preparation, are “nerve at risk periods” during implantation, and should be limited in time and in the amount of tension on the soft tissues to prevent temporary or permanent damage to the axillary nerve, especially when using an adjustable “arm-positioner”, which may prolong permanent tension on the nerve.

Deltoid overlengthening may also cause pain. Schmalzl et al. measured deltoid muscle stiffness using shear wave ultrasound elastography in patients following RSA [45]. Increased tension of the anterior deltoid muscle portion significantly correlated with an increased pain level. 

## 5. Conclusions

ER, EXT, and IR are key challenges to the optimization of RSA function compared to Grammont’s design. They are substantially improved by the combination of glenoid-sided lateralization, inferior GS overhang, and a decreased NSA as low as 135°, as outlined in part I of this review. 

Lateralization, distalization and inferior GS tilt have been proven to be the three most important factors in hierarchical order to increase glenohumeral ABD by increasing the subacromial space. 

Thoracic kyphosis, scapular posture, and motion need to be evaluated, as they may influence planning, positioning, and outcomes of RSA. 

Glenoid-sided lateralization (LGMH) and humeral lateralization (MGLH) have different effects on the moment arm but also muscle length for the force generation of RSA. The total amount of lateralization has been recommended to be close to the anatomical lateralization of the humeral tuberosities; however, the optimal amount of glenoid and humeral contribution is unknown to date.

The ideal RSA configuration most certainly depends on the individual, and patient-specific factors need to be respected and taken into account. Further clinical studies with a controlled design for parameters of interest are needed. This review is a summary of the body of literature on the challenges posed for the optimization of RSA and their interplay. It may be used as an aide memoire, so as not to overlook or compromise on one of the challenges which could lead to unsatisfactory RSA outcomes or complications.

## Figures and Tables

**Figure 1 jcm-12-01616-f001:**
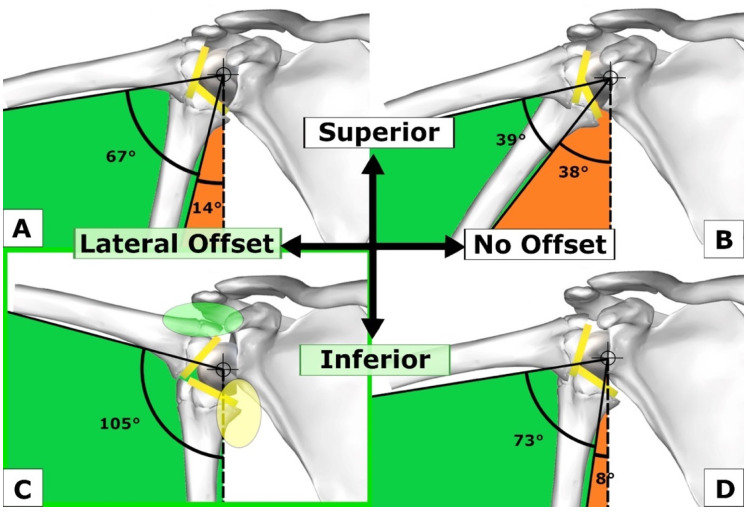
Influence of the position and lateralization of the glenosphere (GS) on abduction. (**A**) Superior GS position with lateral offset; (**B**) Superior GS position without lateral offset; (**C**) Inferior GS position with lateral offset providing the best abduction-adduction range; (**D**) Inferior GS position without lateral offset. Figure reused and modified with permission, Gutiérrez et al. [18].

**Figure 2 jcm-12-01616-f002:**
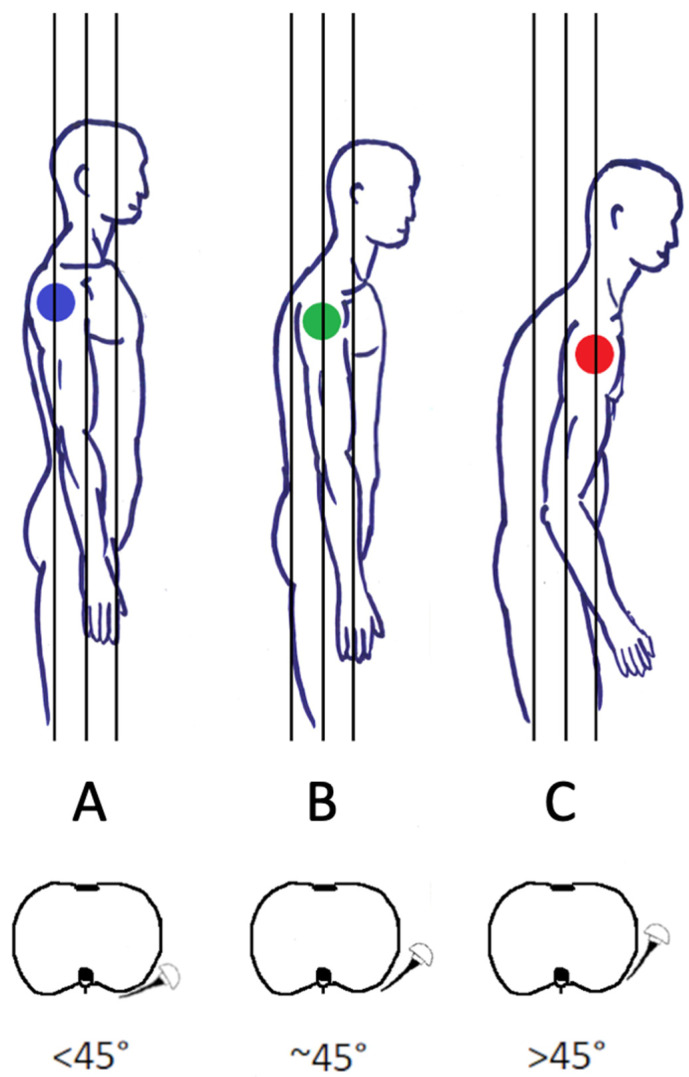
Types of thoracic sagittal balance and scapular protraction and rotation (**A**–**C**). (**A**) Normal scapular protraction and rotation <45°. (**B**) Increased scapular protraction and rotation. (**C**) Severe scapular protraction and rotation >45°. (Figure reused from the article of Bauer et al. [4]. Adapted from the article of Moroder et al. [2].

**Figure 3 jcm-12-01616-f003:**
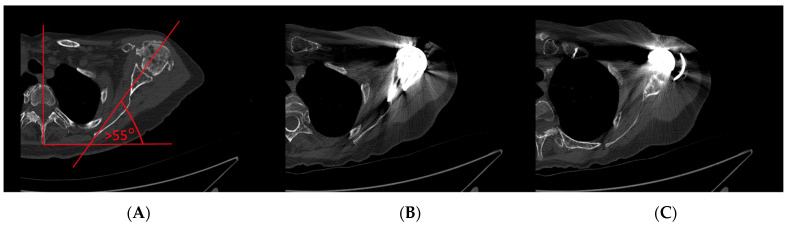
CT images of a clinical case. (**A**) C-type scapular internal rotation and protraction with (**B**) Anterior overstuffing after RSA lateralization and (**C**) Mismatch between the version of the GS, and humeral implant torsion.

**Figure 4 jcm-12-01616-f004:**
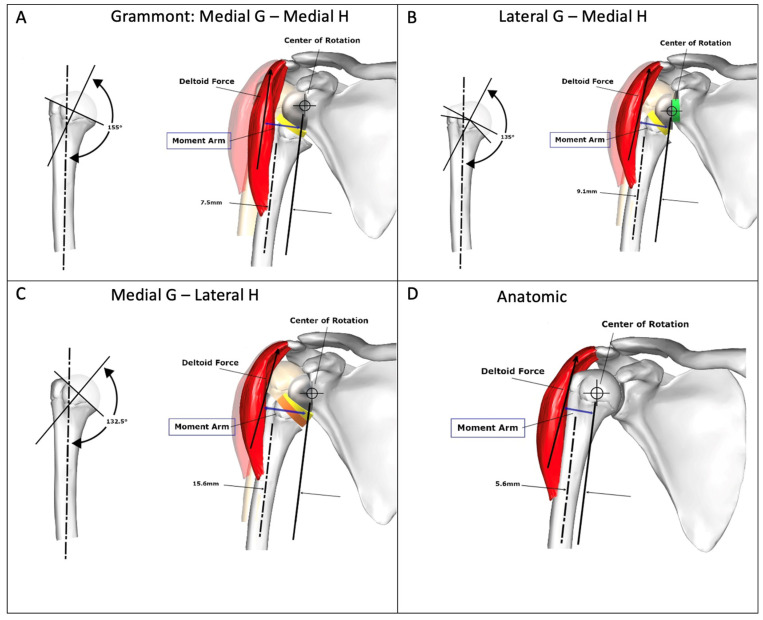
Changes of the length of moment arm (blue) according to RSA design compared to normal anatomy. G: Glenosphere. H: Humerus. (**A**) Medial G—Medial H with increased length of moment arm compared to (**B**) Lateral G—Medial H and (**D**) Anatomic state. (**C**) Medial G—Lateral H provides the longest moment arm. Figure reused and modified with permission, Hamilton et al. [9].

**Figure 5 jcm-12-01616-f005:**
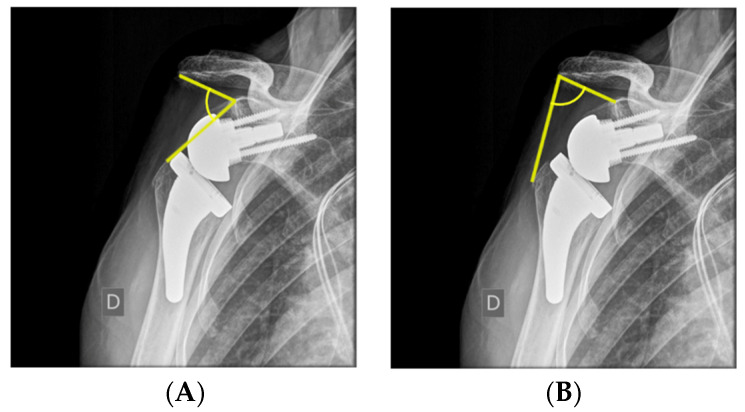
(**A**) Distalization shoulder angle (DSA) defined as the angle between a line connecting the superior glenoid tubercule and the most lateral part of the acromion and a line connecting the superior glenoid tubercule and the most superior part of the greater tuberosity. (**B**) Lateralization shoulder angle (LSA) defined as the angle between a line connecting the superior glenoid tubercule and the most lateral part of the acromion and a line connecting the most lateral part of the acromion and the most lateral part of the greater tuberosity [42].

**Figure 6 jcm-12-01616-f006:**
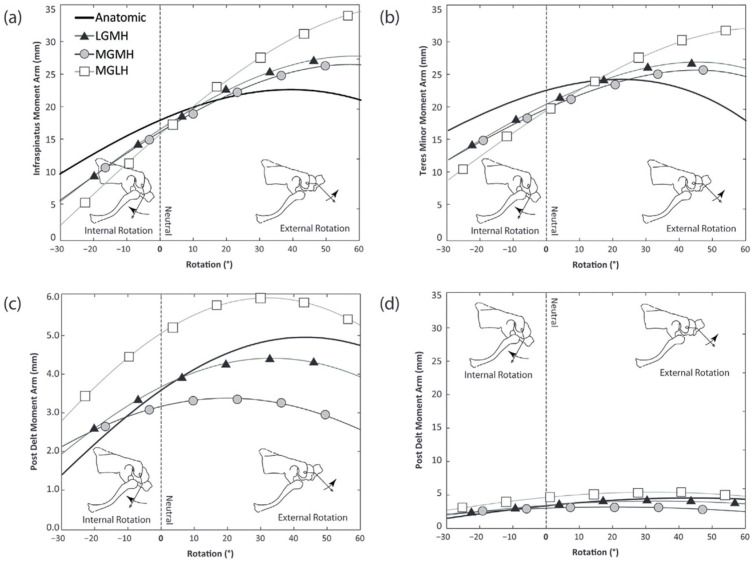
ERO moment arms for rotation from 30° IR to 60° ER at 30° of ABD. Infraspinatus. (**b**) Teres minor. (**c**) Posterior deltoid. (**d**) Same plot of posterior deltoid scaled the same as (**a**,**b**). L: Lateral. M: Medial. G: Glenosphere. H: Humerus. Figure reused with permission, Hamilton et al. [9].

## Data Availability

Not applicable.

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
