# Peer review of "Challenges for Optimization of Reverse Shoulder Arthroplasty Part II: Subacromial Space, Scapular Posture, Moment Arms and Muscle Tensioning"

_jcm, 2023, doi:10.3390/jcm12041616_

Round 1

Reviewer 1 Report

Dear authors,

congratulations to this great overview on influencing geometric variables on RSA.

I enjoyed reading this review and feel many others will too. However, some minor revisions are necessary:

Please recheck for punctuation and spelling.

Line 66. add 'mean' to maximum

Line 69 correct ABD angle.

Line 72 I suggest to delete 'first'

Line 74 add rim : superolateral rim of the insert..

Line 80 delete '2'

Line 90  I suggest to highlight these 4 factors

Line 192 It is the other way round: LAt can be achieved when position the LONG distance of the onlay eccentricity medially.

Reviewer 2 Report

The authors report that the “Lateralization of COR of GS” is one of the most important factors for the Abduction and Adduction of the shoulder. However, if someone see the article from Laderman et al the BIO-RSA results in slight better abduction, better adduction, but worse flexion and worse external rotation in position 1. From the same article it seems to be most important the inferior tilt and the inferior offset of the glenosphere.

Also from the article of the Laderman and the Article from Boutsiadis et al (The lateralization and distalization shoulder angles are important determinants of clinical outcomes in reverse shoulder arthroplasty)  show that the lateralisation from the humeral side is also very important. Please modify.

The subacromial space of the RSA construct is in fact the distalization of the prosthesis. This can be achieved either from the glenoid side with different sizes of the GS ether with different position of it. However, it can be achieved from the humeral side. The prosthesis that distalizes more is the Grammont prosthesis with increased shaft angle. The decrease of this angle can change the position of the RSA. Also the new onlay designs with an eccentric  onlay part can also change the position of the RSA. The latest is shown to the article of Landermann. However, which is the ideal position of the RSA?? Less distalization can lead to worse ABD and more distalization to worse clinical results and neurological problems. The latest is also shown to the article of Boutsiadis et al with LSA angles >70. Please modify.

The authors reported “In contrast to a biomechanical approach to optimize the moment arm for ER, Di Gia- 242 como has presented results of a computer model in January 2022 showing improved del- 243 toid fiber recruitment for ER at 60°of ABD with an RSA lateralized at the glenoid (LGMH) 244 compared to RSA lateralized at the humerus (MGLH)” However, there are a lot of studies that support that the most important is the lateralisation from the humeral side. We do not know which is the best way to lateralise and how not to overlateralize. Overllateralization could lead to opposite results.

Could effort to present a very interesting subject.

MInor changes are necessary.
